# Deep Learning Meets Projective Clustering

**Alaa Maalouf** [*1], **Harry Lang**[*2], **Daniela Rus**[2] **& Dan Feldman**[1]

[1] Robotics & Big Data Labs, Department of Computer Science, University of Haifa
[2] CSAIL, MIT
`alaamalouf12@gmail.com, harry1@mit.edu,`
`rus@csail.mit.edu, dannyf.post@gmail.com`

## Abstract

A common approach for compressing Natural Language Processing (NLP) networks is to encode the embedding layer as a matrix $A \in \mathbb{R}^{n \times d}$, compute its rank-$j$ approximation $A_j$ via SVD (Singular Value Decomposition), and then factor $A_j$ into a pair of matrices that correspond to smaller fully-connected layers to replace the original embedding layer. Geometrically, the rows of $A$ represent points in $\mathbb{R}^d$, and the rows of $A_j$ represent their projections onto the $j$-dimensional subspace that minimizes the sum of squared distances ("errors") to the points. In practice, these rows of $A$ may be spread around $k > 1$ subspaces, so factoring $A$ based on a single subspace may lead to large errors that turn into large drops in accuracy.

Inspired by *projective clustering* from computational geometry, we suggest replacing this subspace by a set of $k$ subspaces, each of dimension $j$, that minimizes the sum of squared distances over every point (row in $A$) to its *closest* subspace. Based on this approach, we provide a novel architecture that replaces the original embedding layer by a set of $k$ small layers that operate in parallel and are then recombined with a single fully-connected layer.

Extensive experimental results on the GLUE benchmark yield networks that are both more accurate and smaller compared to the standard matrix factorization (SVD). For example, we further compress DistilBERT by reducing the size of the embedding layer by $40\%$ while incurring only a $0.5\%$ average drop in accuracy over all nine GLUE tasks, compared to a $2.8\%$ drop using the existing SVD approach. On RoBERTa we achieve $43\%$ compression of the embedding layer with less than a $0.8\%$ average drop in accuracy as compared to a $3\%$ drop previously.

## 1 Introduction and Motivation

Deep Learning revolutionized Machine Learning by improving the accuracy by dozens of percents for fundamental tasks in Natural Language Processing (NLP) through learning representations of a natural language via a deep neural network (Mikolov et al., 2013; Radford et al., 2018; Le and Mikolov, 2014; Peters et al., 2018; Radford et al., 2019). Lately, it was shown that there is no need to train those networks from scratch each time we receive a new task/data, but to fine-tune a full pre-trained model on the specific task (Dai and Le, 2015; Radford et al., 2018; Devlin et al., 2019). However, in many cases, those networks are extremely large compared to classical machine learning models. For example, both BERT (Devlin et al., 2019) and XLNet (Yang et al., 2019) have more than 110 million parameters, and RoBERTa (Liu et al., 2019b) consists of more than 125 million parameters. Such large networks have two main drawbacks: (i) they use too much storage, e.g. memory or disk space, which may be infeasible for small IoT devices, smartphones, or when a personalized network is needed for each user/object/task, and (ii) classification may take too much time, especially for real-time applications such as NLP tasks: speech recognition, translation or speech-to-text.

**Compressed Networks.** To this end, many papers suggested different techniques to compress large NLP networks, e.g., by low-rank factorization (Wang et al., 2019; Lan et al., 2019), prun-

---

[*]equal contribution

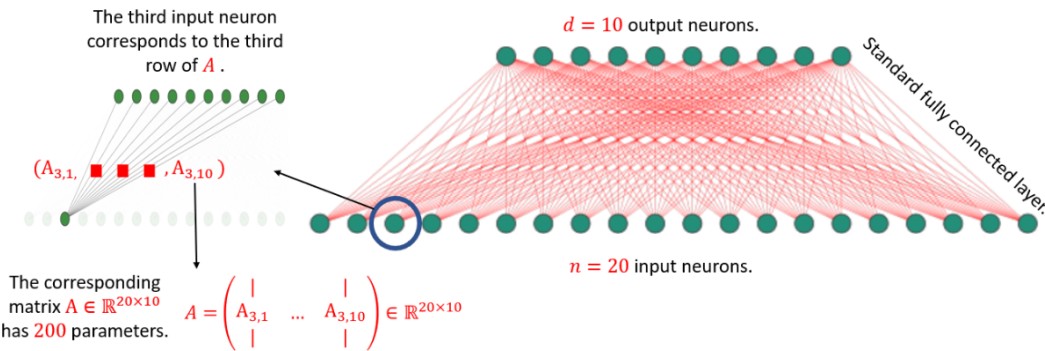

Figure 1: A standard embedding (or fully-connected) layer of 20 input neurons and 10 output neurons. Its corresponding matrix $A \in \mathbb{R}^{20 \times 10}$ has 200 parameters, where the $i$th row in $A$ is the vector of weights of the $i$ neuron in the input layer.

ing (McCarley, 2019; Michel et al., 2019; Fan et al., 2019; Guo et al., 2019; Gordon et al., 2020), quantization (Zafrir et al., 2019; Shen et al., 2020), weight sharing (Lan et al., 2019), and knowledge distillation (Sanh et al., 2019; Tang et al., 2019; Mukherjee and Awadallah, 2019; Liu et al., 2019a; Sun et al., 2019; Jiao et al., 2019); see more example papers and a comparison table in Gordon (2019) for compressing the BERT model. There is no consensus on which approach should be used in what contexts. However, in the context of compressing the embedding layer, the most common approach is low-rank factorization as in Lan et al. (2019), and it may be combined with other techniques such as quantization and pruning.

In this work, we suggest a novel low-rank factorization technique for compressing the embedding layer of a given model. This is motivated by the fact that in many networks, the embedding layer accounts for $20\% - 40\%$ of the network size. Our approach - MESSI: Multiple (parallel) Estimated SVDs for Smaller Intralayers - achieves a better accuracy for the same compression rate compared to the known standard matrix factorization. To present it, we first describe an embedding layer, the known technique for compressing it, and the geometric assumptions underlying this technique. Then, we give our approach followed by geometric intuition, and detailed explanation about the motivation and the architecture changes. Finally, we report our experimental results that demonstrate the strong performance of our technique.

**Embedding Layer.** The embedding layer aims to represent each word from a vocabulary by a real-valued vector that reflects the word's semantic and syntactic information that can be extracted from the language. One can think of the embedding layer as a simple matrix multiplication as follows. The layer receives a standard vector $x \in \mathbb{R}^n$ (a row of the identity matrix, exactly one non-zero entry, usually called *one-hot vector*) that represents a word in the vocabulary, it multiplies $x$ by a matrix $A^T \in \mathbb{R}^{d \times n}$ to obtain the corresponding $d$-dimensional word embedding vector $y = A^T x$, which is the row in $A$ that corresponds to the non-zero entry of $x$. The embedding layer has $n$ input neurons, and the output has $d$ neurons. The $nd$ edges between the input and output neurons define the matrix $A \in \mathbb{R}^{n \times d}$. Here, the entry in the $i$th row and $j$th column of $A$ is the weight of the edge between the $i$th input neuron to the $j$th output neuron; see Figure. 1.

**Compressing by Matrix Factorization.** A common approach for compressing an embedding layer is to compute the $j$-rank approximation $A_j \in \mathbb{R}^{n \times d}$ of the corresponding matrix $A$ via SVD (Singular Value Decomposition; see e.g., Lan et al. (2019); Yu et al. (2017) and Acharya et al. (2019)), factor $A_j$ into two smaller matrices $U \in \mathbb{R}^{n \times j}$ and $V \in \mathbb{R}^{j \times d}$ (i.e. $A_j = UV$), and replace the original embedding layer that corresponds to $A$ by a pair of layers that correspond to $U$ and $V$. The number of parameters is then reduced to $j(n + d)$. Moreover, computing the output takes $O(j(n + d))$ time, compared to the $O(nd)$ time for computing $A^T x$. As above, we continue to use $A_j$ to refer to a rank-$j$ approximation of a matrix $A$.

**Fine tuning.** The layers that correspond to the matrices $U$ and $V$ above are sometimes used only as initial seeds for a training process that is called *fine tuning*. Here, the training data is fed into the

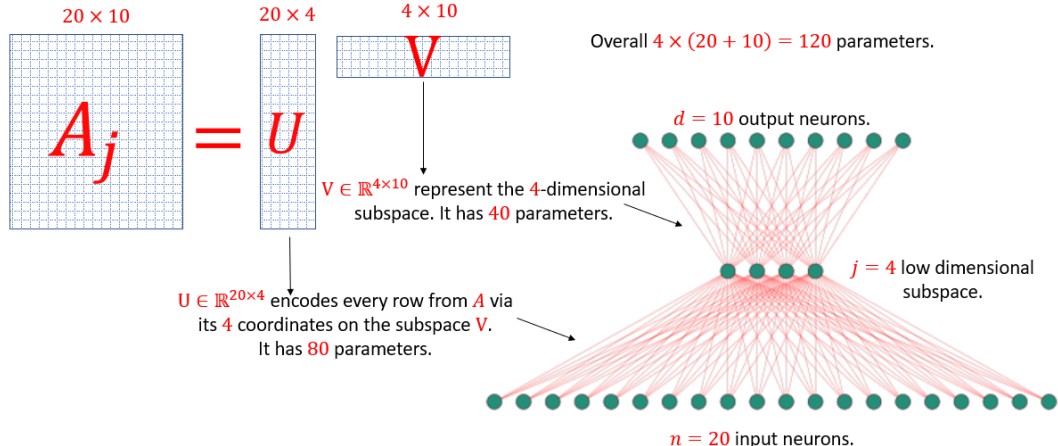

Figure 2: Factorization of the embedding layer (matrix) $A \in \mathbb{R}^{20 \times 10}$ from Figure 1 via standard matrix factorization (SVD) to obtain two smaller layers (matrices) $U \in \mathbb{R}^{20 \times 4}$ and $V \in \mathbb{R}^{4 \times 10}$. In this example, the factorization was done based on a $4$-dimensional subspace. The result is a compressed layer that consists of 120 parameters. The original matrix had 200 parameters. See more details in the figure.

network, and the error is measured with respect to the final classification. Hence, the structure of the data remains the same but the edges are updated in each iteration to give a better accuracy.

Observe that typically, the SVD takes the form $A_j = UD\tilde{V}$, where the columns of $U \in \mathbb{R}^{n \times j}$ are orthogonal, the rows of $\tilde{V} \in \mathbb{R}^{j \times d}$ are orthogonal, and $D \in \mathbb{R}^{j \times j}$ is a diagonal matrix. In this paper and in others, we say that $A_j = UV$ where $V = D\tilde{V}$. Furthermore, the orthogonalization is used only to obtain a low rank approximation $A_j = UV$ using SVD. After that, this property is not kept in the network during the training process (when applying the fine-tuning).

**Geometric intuition.**    The embedding layer can be encoded into a matrix $A \in \mathbb{R}^{n \times d}$ as explained above. Hence, each of the $n$ rows of $A$ corresponds to a point (vector) in $\mathbb{R}^d$, and the $j$-rank approximation $A_j \in \mathbb{R}^{n \times d}$ represents the projection on the $j$-dimensional subspace that minimizes the sum of squared distances ("errors") to the points. Projecting these points onto any $j$-dimensional subspace of $\mathbb{R}^d$ would allow us to encode every point only via its $j$-coordinates on this subspace, and store only $nj$ entries instead of the original $nd$ entries of $A$. This is the matrix $U \in \mathbb{R}^{n \times j}$, where each row encodes the corresponding row in $A$ by its $j$-coordinates on this subspace. The subspace itself can be represented by its basis of $j$ $d$-dimensional vectors ($jd$ entries), which is the column space of a matrix $V^T \in \mathbb{R}^{d \times j}$. Figure 2 illustrates the small pair of layers that corresponds to $U$ and $V$, those layers are a compression for the original big layer that corresponds to $A$.

However, our goal is not only to compress the network or matrix, but also to approximate the original matrix operator $A$. To this end, among all the possible $j$-subspaces of $\mathbb{R}^d$, we may be interested in the $j$-subspace that minimizes the sum of squared distances to the points, i.e., the sum of squared projected errors. This subspace can be computed easily via SVD. The corresponding projections of the rows of $A$ on this subspace are the rows of the $j$-rank matrix $A_j$.

The hidden or statistical assumption in this model is that the rows of the matrix $A$ (that represents the embedding layer) were actually generated by adding i.i.d. Gaussian noise to each point in a set of $n$ points on a $j$-dimensional subspace, that is spanned by what are called latent variables or factors. Given only the resulting matrix $A$, the $j$-subspace that maximizes the likelihood (probability) of generating the original points is spanned by the $j$ largest singular vectors of $A$.

**Why a single distribution?**    Even if we accept the assumption of Gaussian noise, e.g. due to simplicity of computations or the law of large numbers, it is not intuitively clear why we should assume that the rows of $A$ were sampled from a single distribution. Natural questions that arise are:

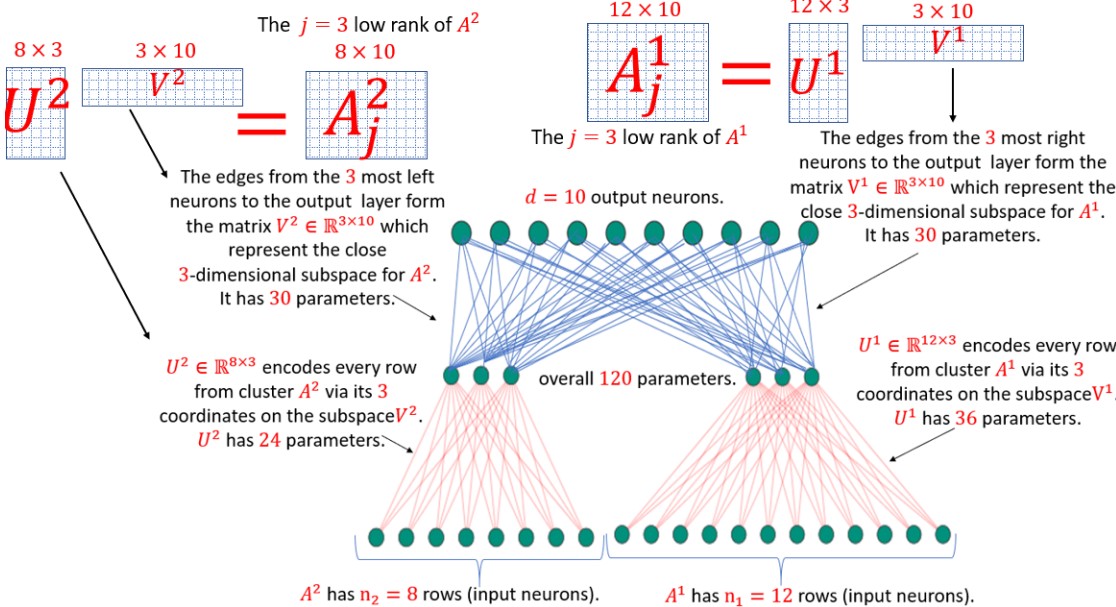

Figure 3: **Example of our compression scheme (MESSI) from A to Z.** Here $j = 3$ and $k = 2$, and we compress the embedding layer from figure 1: (i) find the set of $k = 2$ subspaces, each of dimension $j = 3$, that minimizes the sum of squared distances from each point (row in $A$) to its *closest* subspace. (ii) Partition the rows of $A$ into $k = 2$ different subsets $A^1$ and $A^2$, where two rows are in the same subset if there closest subspace is the same, (iii) for each subset, factor its corresponding matrix into two smaller matrices based on its closest subspace to obtain the $2k = 4$ matrices $U^1, V^1, U^2$ and $V^2$ (where for every $i \in \{1, \cdots, k\}$, the matrix $U^i V^i$ is a low ($j = 3$) rank approximation for $A^i$), (iii) replace the original fully-connected (embedding) layer by 2 layers, where in the first (red color) we have $k = 2$ parallel fully-connected layers for (initialized by) $U^1$ and $U^2$ as in the figure, and the second (blue color) is a fully-connected layer with all the previews $k = 2$, and its weights corresponds to $V^1$ and $V^2$ as follow. For every $i \in \{1, \cdots, k\}$, the weights form the $j = 3$ neurons (nodes) that are connected in the previous layer with $U^i$ are initialized by $V^i$. The result is a compressed layer that consists of $nj + kjd = 20 \times 3 + 2 \times 3 \times 10 = 120$ parameters. See more details in the figure.

  (i) Can we get smaller and/or more accurate models in real-world networks by assuming multiple instead of a single generating distribution (i.e. multiple subspaces)?

  (ii) Can we efficiently compute the corresponding factorizations and represent them as part of a network ?

## 2    OUR CONTRIBUTION

We answer the above open questions by suggesting the following contributions. In short, the answers are:

  (i) In all the real-world networks that we tested, it is almost always better to assume $k \geq 2$ distributions rather than a single one that generated the data. It is better in the sense that the resulting accuracy of the network is better compared to $k = 1$ (SVD) for the same compression rate.

  (ii) While approximating the global minimum is Max-SNP-Hard, our experiments show that we can efficiently compute many local minima and take the smallest one. We then explain how to encode the result back into the network. This is by suggesting a new embedding layer architecture that we call MESSI (Multiple (parallel) Estimated SVDs for Smaller Intralayers); see Figure 3. Extensive experimental results show significant improvement.

**Computational Geometry meets Deep Learning.** Our technique also constructs the matrix $A \in \mathbb{R}^{n \times d}$ from a given embedding layer. However, inspired by the geometric intuition from the previous section, we suggest to approximate the $n$ rows of $A$ by clustering them to $k \geq 2$ subspaces instead of one. More precisely, given an integer $k \geq 1$ we aim to compute a set of $k$ subspaces in $\mathbb{R}^d$, each of dimension $j$, that will minimize the sum over every squared distance of every point (row in $A$) to its nearest subspace. This can be considered as a combination of $j$-rank or $j$-subspace approximation, as defined above, and $k$-means clustering. In the $k$-means clustering problem we wish to approximate $n$ points by $k$ center *points* that minimizes the sum over squared distance between every point to its nearest center. In our case, the $k$ centers points are replaced by $k$ subspaces, each of dimension $j$. In computational geometry, this type of problem is called *projective clustering* (see Figure 4), and its used in many tasks in the fields of Machine Learning and Computer Vision (Feng et al., 2011; Xu et al., 2005; Liu et al., 2012; Trittenbach and Böhm, 2019),

**From Embedding layer to Embedding layers.** The result of the above technique is a set of $k$ matrices $A_j^1, \cdots, A_j^k$, each of rank $j$ and dimension $n_i \times d$ where the $i$th matrix corresponds to the cluster of $n_i$ points that were projected on the $i$th $j$-dimensional subspace. Each of those matrices can be factored into two smaller matrices (due to its low rank), i.e., for every $i \in \{1, \cdots, k\}$, we have $A_j^i = U^i V^i$, where $U^i \in \mathbb{R}^{n_i \times j}$, and $V^i \in \mathbb{R}^{j \times d}$. To plug these matrices as part of the final network instead of the embedded layer, we suggest to encode these matrices via $k$ parallel sub-layers as described in what follows and illustrated in Figure 3.

**Our pipeline: MESSI.** We construct our new architecture as follows. We use $A$ to refer to the $n \times d$ matrix from the embedding layer we seek to compress. The input to our pipeline is the matrix $A$, positive integers $j$ and $k$, and (for the final step) parameters for the fine-tuning.

1. Treating the $n$ rows of $A$ as $n$ points in $\mathbb{R}^d$, compute an approximate $(k, j)$-projective clustering. The result is $k$ subspaces in $\mathbb{R}^d$, each of dimension $j$, that minimize the sum of squared distances from each point (row in $A$) to its *closest* subspace. For the approximation, we compute a local minimum for this problem using the Expectation-Maximization (EM) method (Dempster et al., 1977).

2. Partition the rows of $A$ into $k$ different subsets according to their nearest subspace from the previous step. The result is submatrices $A^1, \ldots, A^k$ where $A^i$ is a $n_i \times d$ matrix and $n_1 + \ldots + n_k = n$.

3. For each matrix $A^i$ where $1 \leq i \leq k$, factor it to two smaller matrices $U^i$ (of dimensions $n_i \times j$) and $V^i$ (of dimensions $j \times d$) such that $U^i V^i$ is the rank-$j$ approximation of $A^i$.

4. In the full network, replace the original fully-connected embedding layer by 2 layers. The first layer is a parallelization of $k$ separate fully-connected layers, where for every $i \in \{1, \cdots, k\}$ the $i$th parallel layer consists of the matrix $U^i$, i.e., it has $n_i$ input neurons and $j$ output neurons. Here, each row of $A$ is mapped appropriately. The second layer is by combining the matrices $V^1, \cdots V^k$. Each of the $k$ output vectors from the previous layer $u_1, \ldots, u_k$ are combined as $V^1 u_1 + \ldots + V^k u_k$; see Figure 3 for an illustration.

5. Fine-tune the network.

The result is a compressed embedding layer. Every matrix $U^i$ has $n_i j$ parameters, and the matrix $V^i$ has $jd$ parameters. Therefore the compressed embedding layer consists of $nj + kjd$ parameters, in comparison to the uncompressed layer of $nd$ parameters.

**Practical Solution.** The projective clustering problem is known to be Max-SNP-hard even for $d = 2$ and $j = 2$, for any approximation factor that is independent of $n$. Instead, we suggest to use an algorithm that provably converges to a local minimum via the Expectation-Maximization (EM) method (Dempster et al., 1977), which is a generalization of the well known Lloyd algorithm (Lloyd, 1982). The resulting clusters and factorizations are used to determine the new architecture and its initial weights; see Figure 3 for more details. We run on instances of AWS Amazon EC2 cloud, and detail our results in the next section.

**Open code and networks.** Complete open code to reproduce the resulting networks is provided. We expect it to be useful for future research, and give the following few examples.

## 2.1 GENERALIZATIONS AND EXTENSIONS.

Our suggested architecture can be generalized and extended to support many other optimization functions that may be relevant for different types of datasets, tasks or applications besides NLP.

$\ell^q$**-error.** For simplicity, our suggested approach aims to minimize sum of *squared* distances to $k$ subspaces. However, it can be easily applied also to sum of distances from the points to the subspace, which is a more robust approach toward outliers ("far away points").

Even for $k = 1$ recent results of Tukan et al. (2020b) show improvement over SVD.

**Distance functions.** Similarly, we can replace the Euclidean $\ell_2$-distance by e.g. the Manhattan distance which is the $\ell_1$-norm between a point $x$ and its projection, i.e., $\|x - x'\|_1$ or sum of differences between the corresponding entries, instead of sum of squared entries, as in the Euclidean distance $\|x - x'\|_2$ in this paper.

**Non-uniform dimensions.** In this paper we assume that $k$ subspaces approximate the input points, and each subspace has dimension exactly $j$, where $j, k \geq 1$ are given integers. A better strategy is to allow each subspace to have a different dimension, $j_i$ for every $i \in \{1, \cdots, k\}$, or add a constraint only on the sum $j_1 + \cdots + j_k$ of dimensions. Similarly, the number $k$ may be tuned as in our experimental results. Using this approach we can improve the accuracy and enjoy the same compression rate.

For more details about those generalizations and others, we refer the interested reader to section E.1 at the appendix.

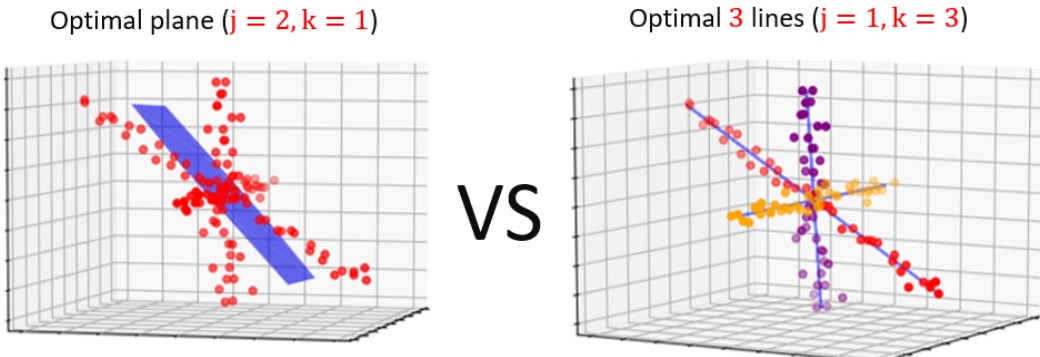

Figure 4: Why $k$ subspaces? Here, we have $n = 120$ data points in $\mathbb{R}^3$ that are spread around $k = 3$ lines ($j = 1$). Factoring this data based on the optimal plane $P$ results with large errors, since some points are far from this plane as can be seen in the left hand side of the figure. On the right hand side, factoring the data based the 3 optimal lines $\ell_1, \ell_2,$ and $\ell_3$ gives a much smaller errors. Also, storing the factorization based on the plane $P$ requires $2(120 + 3) = 246$ parameters, compared to $120 \times 1 + 3 \times 3 = 129$ parameters based on $\ell_1, \ell_2,$ and $\ell_3$. I.e., less memory and a better result.

## 3 EXPERIMENTAL RESULTS

**GLUE benchmark.** We run our experiments on the General Language Understanding Evaluation (GLUE) benchmark (Wang et al., 2018). It is widely-used collection of 9 datasets for evaluating natural language understanding systems.

**Networks.** We use the following networks: (i) RoBERTa (Liu et al., 2019b), it consists of 120 millions parameters, and its embedding layer has 38.9 million parameters (32.5% of the entire network size), (ii) DistilBERT (Sanh et al., 2019) consists of 66 million parameters, and its embedding layer has 23.5 million parameters (35.5% of the entire network size), and (iii) ALBERT (base-v2) (Lan et al., 2019), which consists of 11.7 million parameters, and its embedding layer has 3.8 million parameters (33% of the entire network).

| Model | Embedding layer compression rate | MRPC | COLA | MNLI | SST-2 | STS-B | QNLI | RTE | WNLI | QQP | Avg. |
|---|---|---|---|---|---|---|---|---|---|---|---|
| DistilBERT | 20% 
 $k$ and $j$ | 0.98 
 4 \| 558 | 3.4 
 4 \| 558 | −0.44 
 4 \| 558 | −0.54 
 4 \| 558 | −0.23 
 4 \| 558 | −0.84 
 5 \| 545 | −1.08 
 7 \| 522 | 0 
 5 \| 545 | 0.26 
 5 \| 545 | 0.17 |
| DistilBERT | 40% 
 $k$ and $j$ | 1.2 
 6 \| 400 | 3.7 
 5 \| 409 | −0.1 
 4 \| 418 | 0.9 
 5 \| 409 | −0.15 
 5 \| 409 | −0.7 
 3 \| 428 | −0.72 
 7 \| 392 | 0 
 5 \| 409 | 0.4 
 5 \| 409 | 0.5 |
| DistilBERT | 50% 
 $k$ and $j$ | 3.1 
 6 \| 333 | 8.6 
 5 \| 341 | −0.27 
 5 \| 341 | 1.4 
 5 \| 341 | −0.4 
 5 \| 341 | NA | 0.36 
 7 \| 326 | 0 
 5 \| 341 | NA | 1.8 |
| RoBERTA | 25% − 35% 
 $k$ and $j$ | 0.2 
 5 \| 517 | 2.3 
 10 \| 451 | 0.2 
 5 \| 517 | −0.5 
 5 \| 517 | 0.44 
 5 \| 517 | 0.2 
 5 \| 517 | 1 
 10 \| 451 | 0 
 5 \| 517 | NA | 0.47 |
| RoBERTA | 40% − 50% 
 $k$ and $j$ | 0.2 
 5 \| 384 | 3.4 
 10 \| 384 | 0.61 
 5 \| 384 | −0.3 
 5 \| 384 | 0.63 
 5 \| 384 | 0.2 
 5 \| 384 | 1 
 10 \| 384 | 0 
 5 \| 384 | NA | 0.71 |

Table 1: In the table above, we present the compressed models with the best accuracy achieved for specific compression rates (or intervals) of the embedding layer. We report their drop in accuracy, and the used values of $k$ and $j$. Specifically, in each entry the "accuracy drop" is presented above the used $k$ and $j$ values. The last column is the average accuracy drop over all tested tasks. Observe that: (i) negative values presents improvements in the accuracy upon the non-compressed version of the corresponding model, and (ii) the results in this table can be improved if we allow to use the best model from higher compression rates also, e.g., in the task RTE on the network DistilBERT, we achieved 2.5 accuracy increase when we compressed 60% of the embedding layer, however, in this table we did not add this result for the smaller compression rates of 20, 40 and 50.

**Software and Hardware.** All the experiments were conducted on a AWS c5a.16xlarge machine with 64 CPUs and 128 RAM [GiB]. To build and train networks, we used the suggested implementation at the Transformers [1] library from HuggingFace (Wolf et al., 2019) (Transformers version 3.1.0, and PyTorch version 1.6.0 (Paszke et al., 2017)). For more detailes about the implementation, we refer the reader to section A at the appendix.

**The setup.** All our experiments are benchmarked against their publicly available implementations of the DistilBERT, RoBERTa, and ALBERT models, fine-tuned for each task, which was in some cases higher and in other cases lower than the values printed in the publications introducing these models. Given an embedding layer from a network that is trained on a task from GLUE, an integer $k \geq 1$, and an integer $j \geq 1$. We build and initialize a new architecture that replaces the original embedding layer by two smaller layers as explained in Figure 3. We then fine tune the resulted network for 2 epochs. We ran the same experiments for several values of $k$ and $j$ that defines different compression rates. We compete with the standard matrix factorization approach in all experiments.

## 3.1 REPORTED RESULTS

**Compressing RoBERTA and DistilBERT.** (i) In Figures 5 and 6 the $x$-axis is the compression rate of the embedding layer, i.e. a compression of 40% means the layer is 60% its original size. The $y$-axis is the accuracy drop (relative error) with respect to the original accuracy of the network (with fine tuning for 2 epochs). In Figure 5, each graph reports the results for a specific task from the GLUE benchmark on RoBERTa, while Figure 6 reports the results of DistilBERT.

(ii) On the task WNLI we achieved 0 error on both networks using the two approaches of SVD and our approach until 60% compression rate, so we did not add a figure on it.

(iii) In RoBERTa, we checked only 2 compression rates on MNLI due to time constraints, and we achieved similar results in both techniques, e.g., we compressed 45% of the embedding layer, based on our technique with $k = 5$ and $j = 384$ to obtain only 0.61% drop in accuracy with fine tuning and 4.2% without, this is compared to 0.61% and 13.9% respectively for the same compression rate via SVD factorization. In DistilBERT, we compressed 40% of the embedding layer with $k = 4$ and achieved a 0.1% *increase* in accuracy after fine-tuning, as compared to a 0.05% drop via SVD factorization (on MNLI).

(iv) Table 1 suggests the best compressed networks in terms of accuracy VS size.

---

[1]https://github.com/huggingface/transformers

| Model | Parameters | MRPC | COLA | MNLI | SST-2 | STS-B | RTE | WNLI | QQP |
|---|---|---|---|---|---|---|---|---|---|
| ALBERT (base-v2) | 11.7M | 89.7 | 57.7 | 84.9 | 92.5 | 90.5 | 77.6 | 59.2 | 90.7 |
| MESSI-ALBERT (base-v2) | 11.7M | 90 | 58.5 | 84.7 | 92.7 | 90.5 | 78.3 | 60.6 | 90.7 |

Table 2: In the table above, we report the accuracy achieved by "ALBERT (base-v2)" model on the tasks from GLUE, and we compare them to the results achieved on another model of the same size (up to $0.18\%$ increase) which we call "MESSI-ALBERT (base-v2)". This model is exactly the same as the original "ALBERT (base-v2)" model up to one change, where the original embedding layer of "ALBERT (base-v2)" (consists of 30k rows and 128 columns) is modified to the new suggested MESSI architecture, with $k = 7$, and $j = 125$ according the pipeline at Section 2, and without fine-tuning. It can be seen by the table, that the new architecture achieved a better results.

**Improving the accuracy of pre-trained models using MESSI.** In Table 2, we test if the MESSI architecture can improve the accuracy of a pre-trained model, while maintaining the same number of parameters. The only change done on the given model is factoring its embedding layer to the suggested architecture using the detailed pipeline at section 2. Here, we make sure to choose the right values of $k$ and $j$ such that the original embedding layer size is maintained (up to a very small change). We conducted this experiment on the model ALBERT (base v2). The results are actually promising.

**More results that are placed in the appendix:** (i) Figure 8 in section B shows the accuracy drop as a function of the compression rate on the RoBERTA model before fine-tuning. (ii) In section C we compress a fully-connected layer in different settings, specifically speaking we compress the two popular models: LeNet-300-100 on MNIST (LeCun et al., 1998), and VGG-19 (Simonyan and Zisserman, 2014) on CIFAR10 (Krizhevsky et al., 2009), see results at Figures 9 and 10. (iii) In section D, we suggest a way to determine the values of $k$ and $j$ in practice for a given compression rate, and we report the results on compressing DistilBERT based on this suggestion; see Figure 11. (iv) Finally, in section E we check how another clustering method can fit in our pipeline, i.e., instead of clustering the input neurons of the fully-connected layer (rows of $A$) via projective clustering (steps 1 and 2 in the pipeline at Section 2), we try the known $k$-means clustering, and then we continue the same by applying SVD on each cluster and building the corresponding new layers. See results in Figures 12, 13, 14 and 15.

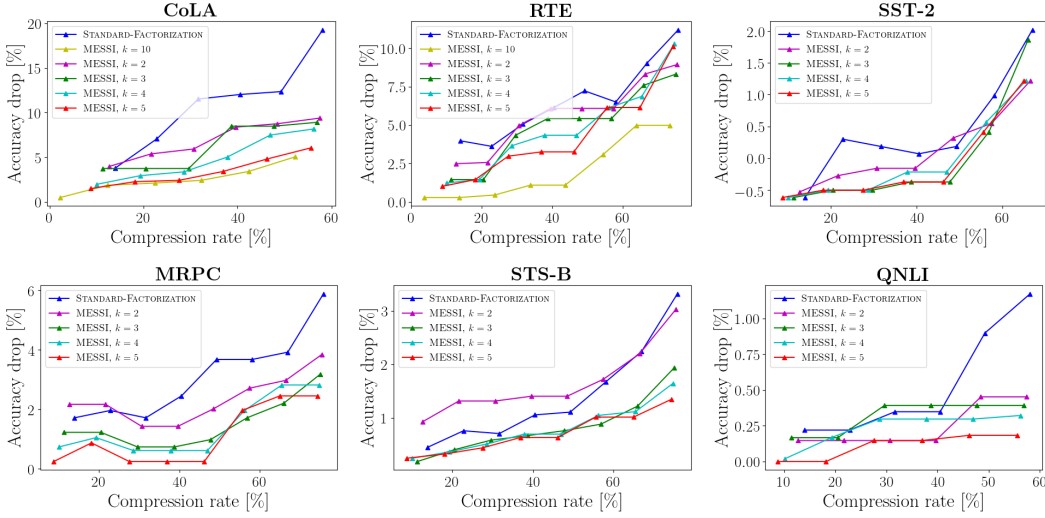

Figure 5: Results on RoBERTa: Accuracy drop as a function of compression rate, with fine tuning for 2 epochs after compression. To illustrate the dependence of MESSI on the choice of $k$, we have plotted several contours for constant-$k$. As the reader will notice, the same dataset may be ideally handled by different values of $k$ depending on the desired compression.

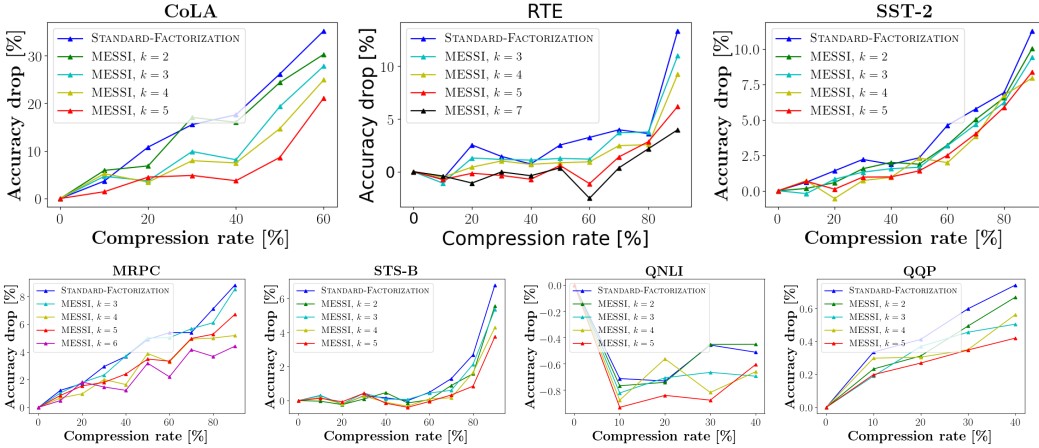

Figure 6: Results on DistilBERT: Accuracy drop as a function of compression rate, with fine tuning for 2 epochs after compression.

## 3.2 DISCUSSION

As shown by Figures 5 and 6, our approach outperforms the traditional SVD factorization. In all experiments, our method achieves better accuracy for the same compression rate compared to the traditional SVD. For example, in RobERTa, we compress $43\%$ of the embedding layer with less that $0.8\%$ average drop in accuracy, this is compared to the $3\%$ drop in the standard technique for a smaller compression rate of $40\%$. In DistilBERT, we achieved $40\%$ compression of the embedding layer while incurring only a $0.5\%$ average drop in accuracy over all nine GLUE tasks, compared to a $2.8\%$ drop using the existing SVD approach. As the reader will notice, the same dataset (and network) may be ideally handled by different values of $k$ depending on the desired compression.

We observed that our technique shines mainly when the network is efficient, and any small change will lead to large error, e.g., as in the CoLA/RTE/MRPC graph of Figure 5. Although we achieve better results in all of the cases, but here the difference is more significant (up to $10\%$), since our compressed layer approximates the original layer better than SVD, the errors are smaller, and the accuracy is better. Furthermore, Figure 8 shows clearly that even without fine tuning, the new approach yields more accurate networks. Hence, we can fine tune for smaller number of epochs and achieve higher accuracy. Finally, by Table 2 we can see that the MESSI architecture can be used also to improve the accuracy of pre-trained models while maintaining the original size.

## 3.3 CONCLUSION

We suggested a novel approach for compressing a fully-connected layer. This is by clustering the input neurons of the layer into $k$-subsets (via projective clustering) and then factoring the corresponding weights matrix of each subset. We then provided a novel architecture that replaces the original fully-connected layer by a set of $k$ small layers that operate in parallel and are then recombined with a single fully-connected layer. The experimental results showed that our suggested algorithm overcomes the traditional factorization technique and achieves higher accuracy for the same compression rate before and after fine-tuning.

## 3.4 FUTURE WORK

The future work includes experiments on other networks and data sets both from the field of NLP and outside it, e.g., an inserting experiment is to modify the ALBERT network (Lan et al., 2019), by changing its embedding layer architecture (that consists of two layers based on the standard matrix factorization) to the suggested architecture in this paper, while maintaining the same number of parameters, and to check if this modification improved its accuracy, also the suggested generalizations and extensions from section 2.1 should be tried, where we strongly believe they will allow us to achieve even better results. Finally, generalizing the approach to other type of layers.

## 4    Acknowledgements

Support for this research has been provided in part by NSF award 1723943. We are grateful for it.

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

## A    IMPLEMENTATION IN PYTORCH

Since we did not find straight support for the new suggested architecture, we implemented it as follows. To represent the matrices $V^1, \cdots, V^k$ that are described is Section 2, we concatenate them all to a one large matrix $V = [(V^1)^T, \cdots, (V^k)^T]^T$ of $kj$ rows and $d$ columns, and we build a fully-connected layer the corresponds to $V$. For the $k$ parallel layers (matrices) $U^1, \cdots, U^k$, we build one large sparse matrix $U$ of $n$ rows and $kj$ columns. Every row of this matrix has at least $(k-1)j$ zero entries, and at most $j$ non zero entries, where the non-zero entries of the $i$th row corresponds to the rows in matrix $V$ which encode the closest subspace to that row's point.

Finally, during the fine tuning or training, we set those zero entries in $U$ as non-trainable parameters, and we make sure that after every batch of back-propagation they remain zero. Hence we have at most $nj$ non-zero entries (trainable parameters) in $U$ and $nj + ndk$ in total.

We hope that in the future, the suggested architecture will be implemented in the known Deep-Learning libraries so it can be easily used while taking advantage of the substantial time and space benefits presented in this paper.

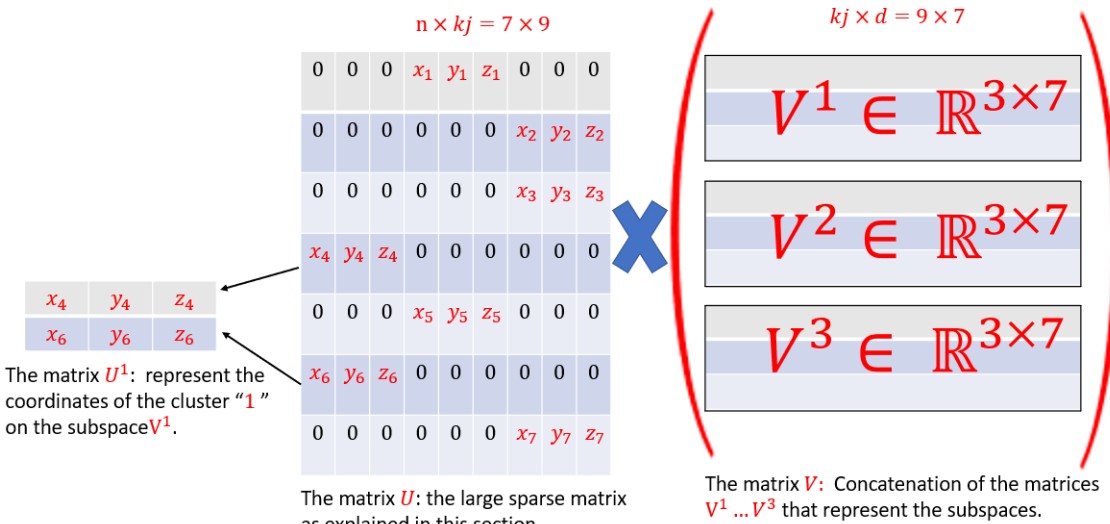

Figure 7: **Implementation.** Example of the factorization $A = UV$ in our implementation. Here $n = 7$ and $d = 7$. The matrix $U$ is built such that row $z$ contains a row from $U^i$ where point $z$ was partitioned to the $i^{\text{th}}$ subspace. In this example, the $4^{\text{th}}$ and $6^{\text{th}}$ rows were both clustered to the first subspace. Hence, the first 3 coordinates of the corresponding rows in the representation matrix $U$ are nonzero, and the other entries are zero. In this way, we used $jk$ dimensions so that none of the $k$ subspaces of dimension $j$ interact.

# B    RESULTS BEFORE FINE TUNING

In this section we report the result of compressing RoBERTa without fine-tuning. By Figure 8 we can clearly see that even without fine tuning, the new approach yields more accurate networks compared to the standard SVD factorization. Hence, our approach gives a better start for the learning (fine-tuning) process, which implies that we can fine tune for smaller number of epochs and achieve higher accuracy and smaller networks.

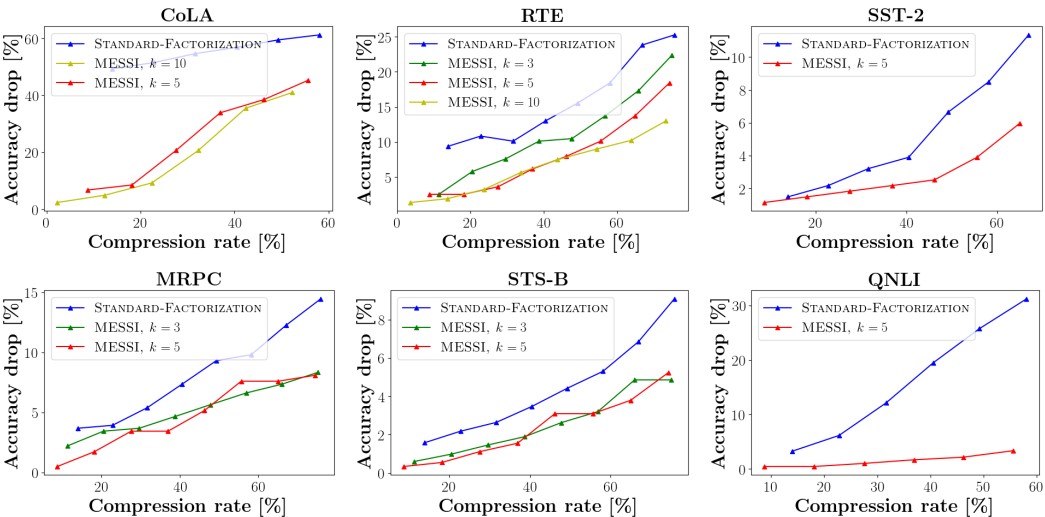

Figure 8: Compressing RoBERTa results: Accuracy drop as a function of compression rate, without fine tuning.

## C   COMPRESSING FULLY-CONNECTED LAYERS USING MESSI.

In this section we test our approach on two popular models: LeNet-300-100 on MNIST (LeCun et al., 1998), and VGG-19 (Simonyan and Zisserman, 2014) on CIFAR10 (Krizhevsky et al., 2009). Also here, we conducted our experiments on the same hardware described in Section 3.

In both experiments, we test our approach on multiple values of $k$ and compare it to $k = 1$ (standard SVD factorization). For every value of $k$, we compress each layer from the hidden fully-connected layers of the given model by the same percentage and using the same value of $k$.

**LeNet-**300-100**.**   The network consists of 266610 parameters, and it is comprised of two fully-connected hidden layers with 300 and 100 neurons, respectively, trained on the MNIST data set.

We test our approach on $k \in \{2, 3, 4, 5\}$. In Figure 9, we report the accuracy drop as a function of the compression rate for the whole network. We can see the advantage of our approach when compressing more than 90% of the network.

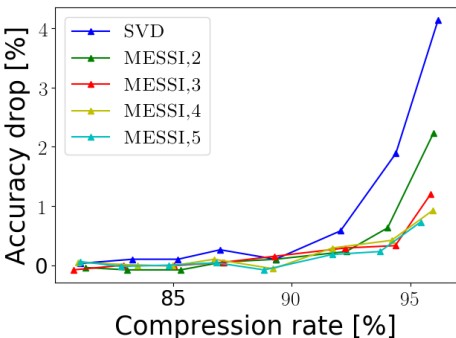

Figure 9: Compressing LeNet-300-100: Accuracy drop as a function of compression rate

**VGG-**19**.**   We used the implementation at [2]. The network consists of 16 convolutional layers, followed by 2 dense hidden (fully-connected) layers with 512 neurons each. Finally, the classification layer has 10 neurons. The fully-connected layers consists of 530442 parameters.

Here, we tested our approach for $k \in \{2, 5\}$. In Figure 10, we report the accuracy drop as a function of the compression rate of the fully-connected layers. The suggested approach has a clear advantage for high compression rates.

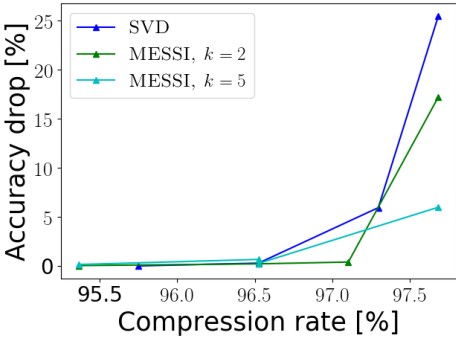

Figure 10: Compressing VGG-19: Accuracy drop as a function of compression rate of the fully-connected layers in the network

---

[2]https://github.com/chengyangfu/pytorch-vgg-cifar10/blob/master/vgg.py

# D    MESSI-ENSEMBLE

In this section we show only the best computed results of DistilBERT: that is obtained by training models at several $k$ values and then evaluating the model that achieves the best accuracy on the training set. Specifically, given a fully-connected layer of $n$ input neurons and $d$ output neurons, for a given compression rate $x$ (e.g., $x = 0.4$ means that we want to remove $40\%$ of the parameters), we try multiple values of $k$ via binary search on $k$. For every such $k$ value we compute the implied value $j = (1-x)dn/(n+kd)$, and we compress the network based on those $k$ and $j$ via the MESSI pipeline. Finally, we save the model that achieves the best accuracy on the training set, and evaluate its results on the test set. Figure 11 reports the results for this approach.

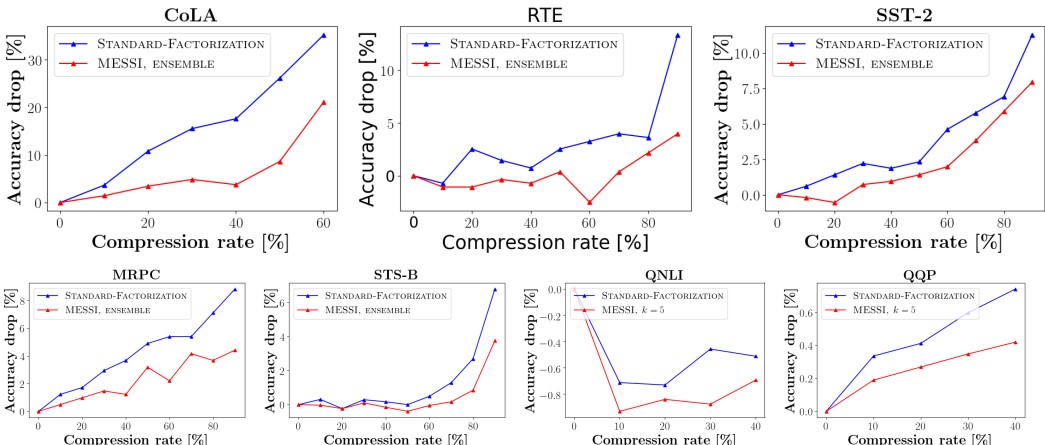

Figure 11: Results on DistilBERT: Accuracy drop as a function of compression rate, with fine tuning for 2 epochs after compression. The red line (MESSI, ensemble) is obtained by training models at several $k$ values and then evaluating the model that achieves the best accuracy on the training set.

# E    PROTECTIVE CLUSTERING VS $k$-MEANS

Recall the suggested pipeline from section 2: The first step of it is to compute a set of $k$ subspaces in $\mathbb{R}^d$, each of dimension $j$ that approximates the $(k,j)$-projective clustering of the input matrix $A$. Then, the second step partitions the input neurons (rows of $A$) according to their closest subspace from the set of $k$ subspace that is computed in the first step. Then, in step 3, we compute the SVD for each cluster, and in steps 4 and 5 we build (and possibly fine-tune) the corresponding architecture as described (see in Figure 3).

In this section, we compare using projective clustering to using $k$-means clustering. We do not apply steps 1 and 2, as we instead partition the input neurons (rows of $A$) into $k$ groups via applying $k$-means clustering on them (instead of projective clustering). We then apply steps 3, 4 and 5 in exactly the same way.

Here, we evaluated our results on the networks: RoBERTa (Liu et al., 2019b) and DistilBERT (Sanh et al., 2019) on the RTE and MRPC tasks from the GLUE benchmark (Wang et al., 2018). Figures 12 and 13 compare the results on RoBERTA between the two clustering methods, with and without fine-tuning, respectively, while Figures 14 and 15 do the same for the results on DistilBERT.

We also used the LeNet-300-100 model on MNIST LeCun et al. (1998) to check this (same) experiment in a different setting. See Figure 16.

**Discussion.**    In Figure 13, where we test the accuracy drop before fine-tuning, we can see that using projective clustering for partitioning the neurons is better than running $k$-means on them, i.e., the projective clustering approach yielded a better start (accuracy before fine-tuning) than the $k$-means approach for the learning process.

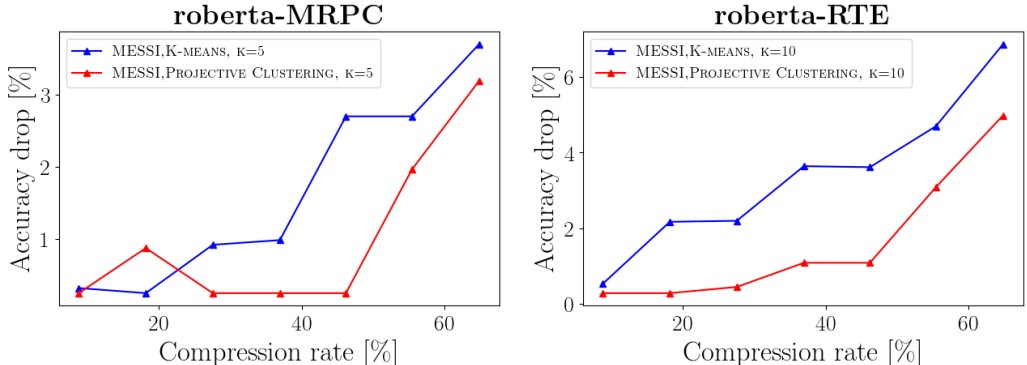

Figure 12: Compressing RoBERTa (with two epochs of fine-tuning): Accuracy drop as a function of compression rate, comparing the projective clustering approach to the known $k$-means clustering.

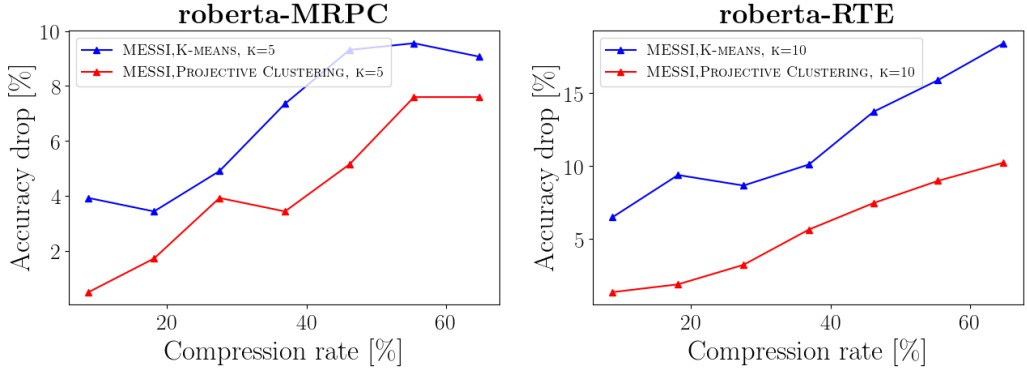

Figure 13: Compressing RoBERTa (without fine-tuning): Accuracy drop as a function of compression rate, comparing the projective clustering approach to the known $k$-means clustering.

This could be explained by the fact that our original approach (projective clustering) aims to compute a set of $k$ subspaces (each of dimension $j$) that minimizes the sum of squared distances from each row in the input matrix $A$ (neuron) to its closest subspace from the set. Hence, factoring the matrix $A$ based on those subspaces gives a good approximation for it, which is not the case in the $k$-means clustering.

This advantage may explain the difference between the two approaches after fine-tuning for the same number of epochs as can be seen in Figure 12.

On the other hand, in Figure 15, the two methods gave similar results in terms of accuracy before fine tuning, and we can see that this effects the results after the fine-tuning, where the two approaches also succeeded to get similar results as can be seen in Figure 13.

Hence, the better way to determine the partition (which determines the compressed architecture) and to initialize the new layer in the MESSI pipeline is the projective clustering approach.

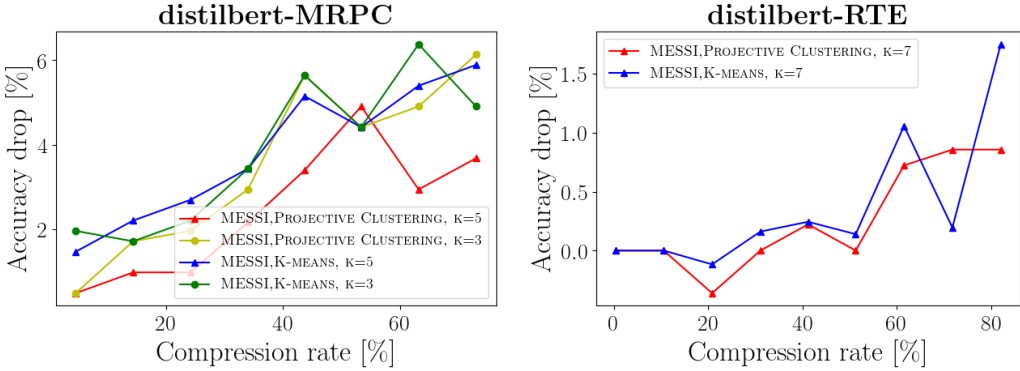

Figure 14: Compressing DistilBERT (with two epochs of fine-tuning): Accuracy drop as a function of compression rate, comparing the projective clustering approach to the known $k$-means clustering.

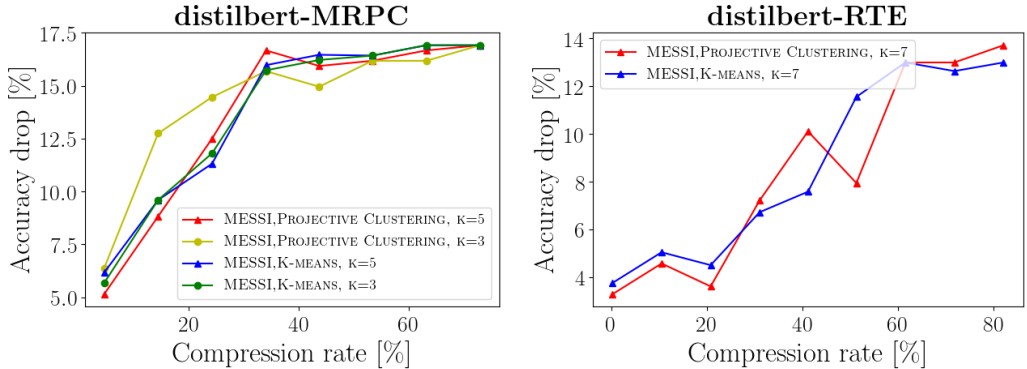

Figure 15: Compressing DistilBERT (without fine-tuning): Accuracy drop as a function of compression rate, comparing the projective clustering approach to the known $k$-means clustering.

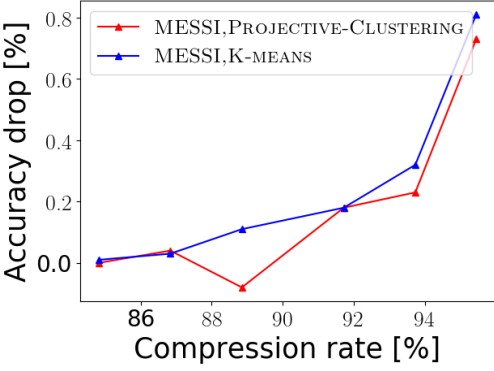

Figure 16: Compressing LeNet-300-100: Accuracy drop as a function of compression rate. Here we compare the projective clustering approach to the known $k$-means clustering.

### E.1 GENERALIZATIONS AND EXTENSIONS.

Here, we give more details about the suggested generalizations and extensions from section 2.1, we also add few more:

$\ell^q$**-error.** For simplicity, our suggested approach aims to minimize sum of *squared* distances to $k$ subspaces. However, it can be easily applied also to sum of distances from the points to the subspace. In this case, we aim to compute the maximum-likelihood of the generating subspaces assuming a Laplacian instead of Gaussian distribution. More generally, we may want to minimize the sum over every distance to the power of $q > 0$., i.e., we take the $q$-norm $\|err\|_q$ where $err$ is the distance between a point to its projection on its closest subspace.

Even for $k = 1$ recent results of Tukan et al. (2020b) show improvement over SVD.

Observe that given the optimal subspaces, the system architecture in these cases remains the same as ours in Figure 3.

**Distance functions.** Similarly, we can replace the Euclidean $\ell_2$-distance by e.g. the Manhattan distance which is the $\ell_1$-norm between a point $x$ and its projection, i.e., $\|x - x'\|_1$ or sum of differences between the corresponding entries, instead of sum of squared entries, as in the Euclidean distance $\|x - x'\|_2$ in this paper. More generally, we may use the $\ell_p$ distance $\|x - x'\|_p$, or even non-distance functions such as M-Estimators that can handle outliers (as in Tukan et al. (2020a)) by replacing $\mathrm{dist}(p, x)$ with $\min\{\mathrm{dist}(p, x), t\}$ where $t > 0$ is constant (threshold) that makes sure that far away points will not affect the overall sum too much.

From an implementation perspective, the EM-algorithm for $k$-subspaces uses a $k = 1$ solver routine as a blackbox. Therefore extending to other distance functions is as simple as replacing the SVD solver (the $k = 1$ for Euclidean distance) by the corresponding solver for $k = 1$.

**Non-uniform dimensions.** In this paper we assume that $k$ subspaces approximate the input points, and each subspace has dimension exactly $j$, where $j, k \geq 1$ are given integers. A better strategy is to allow each subspace to have a different dimension, $j_i$ for every $i \in \{1, \cdots, k\}$, or add a constraint only on the sum $j_1 + \cdots + j_k$ of dimensions. Similarly, the number $k$ may be tuned as in our experimental results. Using this approach we can improve the accuracy and enjoy the same compression rate. This search or parameter tuning, however, might increase the computation time of the compressed network. It also implies layers of different sizes (for each subspace) in Figure 3.

**Dictionary Learning.** Our approach of projective clustering is strongly related to Dictionary Learning (Tosic and Frossard, 2011; Mairal et al., 2009). Here, the input is a matrix $A \in \mathbb{R}^{n \times d}$ and the output is a "dictionary" $V^T \in \mathbb{R}^{d \times j}$ and projections or atoms which are the rows of $U \in \mathbb{R}^{n \times j}$ that minimize $\|A - UV\|$ under some norm. It is easy to prove that $UV$ is simply the $j$-rank approximation of $A$, as explained in Section 1. However, if we have additional constraints, such as that every row of $U$ should have, say, only $k = 1$ non-zero entries, then geometrically the columns of $V^T$ are the $j$ lines that intersects the origin and minimize the sum of distances to the points. For $k > 1$ every point is projected onto the subspace that minimizes its distance and is spanned by $k$ columns of $V^T$.

**Coresets.** Coresets are a useful tool, especially in projective clustering, to reduce the size of the input (compress it in some sense) while preserving the optimal solution or even the sum of distances to any set of $k$ subspaces. However, we are not aware of any efficient implementations and the dependency on $d$ and $k$ is usually exponential as in Edwards and Varadarajan (2005). A natural open problem is to compute more efficient and practical coresets for projective clustering.

### E.2 EXPERIMENTING ON $\ell^q$-ERROR

To get a taste of the suggested extensions, we tried the first suggestion of $\ell^q$-error, with $q = 1$. I.e., we cluster the rows of the input matrix $A$ based on the set of $k$-subspaces that minimizes the sum of (non-squared) distances from each row in $A$ to its closest subspace from the set.

The local minimum of the new clustering problem can still be obtained by the suggested EM algorithm. The only difference is that the SVD computation of the optimal subspace for a cluster of points ($k = 1$) should be replaced by more involved approximation algorithm for computing the

| Clustering method | Task | Fine-tuning epochs | $j = 517$ | $j = 348$ |
|:---:|:---:|:---:|:---:|:---:|
| PC-$\ell^1$ | MRPC | 0 | 3.1 | 5.1 |
| PC-$\ell^2$ | MRPC | 0 | 3.9 | 5.1 |
| PC-$\ell^1$ | MRPC | 2 | 0.2 | 0.2 |
| PC-$\ell^2$ | MRPC | 2 | 0.2 | 0.2 |
| PC-$\ell^1$ | RTE | 0 | 5 | 6.8 |
| PC-$\ell^2$ | RTE | 0 | 3.2 | 7.4 |
| PC-$\ell^1$ | RTE | 2 | 0.47 | 1.2 |
| PC-$\ell^2$ | RTE | 2 | 0.44 | 1 |

Table 3: In the table above, we compare two approaches to cluster the rows of the input matrix $A$ (and check how they fit in the MESSI pipeline), the first approach is projective clustering with $\ell^1$ error (PC-$\ell^1$), and the second is the standard protective clustering with $\ell^2$ error that we used in all the other experiments.

subspace that minimizes sum over distances to the power of $q = 1$; see e.g. Tukan et al. (2020b); Clarkson and Woodruff (2015).

However, this change increased the running time of the algorithm from minutes to days, this is due to the fact the deterministic approximation algorithms for the new problem ($\ell^1$-error) with $k = 1$ take a time of $O(nd^4)$ at least, where $d = 768$ in our case, and we need to run this approximation algorithm many times in the EM procedure. For that, we conducted our experiments only on one network (RoBERTA) on 2 tasks from the GLUE benchmark (MRPC and RTE).

Table 3, shows the accuracy drop for both techniques for two values of $j$ with $k = 5$ on the MRPC task, and the same on RTE with $k = 10$. It can be seen from the table, that mostly, using the $\ell^1$ error as an initialization is better than the $\ell^2$. However,for some reason (that needs further investigation) after fine-tuning for 2 epochs both approaches reached almost the same accuracy, even more, the $\ell^2$ approach achieved a better accuracy sometime. We leave this for future research.

