# OpenReview forum: "Deep Learning meets Projective Clustering"
_ICLR.cc/2021/Conference — ICLR 2021 Poster_

### Official Review · AnonReviewer4 · 2020-10-28
**Nice application of projective clustering to model compression**

**Rating:** 7
**Confidence:** 4

**Review:**

Summary:
This paper applies projective clustering to the embedding layer of deep networks with large model sizes such as RoBERTa. The idea of finding more than one subspaces to factorize the embedding weight matrix has nice intuition and insights. I vote for accepting.

Strengths:
1. The paper has convincing evidence showing the reduction in percent of accuracy drop when applying projective clustering to the embedding weight vectors.
2. The paper has illustration figures that clearly show the intuition of the approach as well as how the compression is achieved.

Weaknesses:
1. It would be better if more baselines can be included in the experiment comparisons. In particular, Since Step 2-3 of the proposed MESSI pipeline (page 4) is partitioning of all the input neurons and computing SVD for each partition, I would be really interested in seeing the comparison of projective clustering vs simpler clustering methods such as k-means to partition the input neurons, in the evaluation.
2. The authors discussed extensions such as using $L_1$ error and $L_1$ distance, but no experiments were performed for the extensions. Some experiment results will be better to establish the flexibility of the framework of projective clustering in model compression tasks.

Questions during rebuttal period:
1. Please provide some results regarding the weaknesses above, especially the result of more baseline methods.
2. Is projective clustering the only way to find clusters in multiple subspaces? What are some alternatives? For example, in subspace clustering, all the data points can be projected to the same subspace and form clusters; we may run subspace clustering for multiple times to get clustering results in different subspaces.

---

> ### Author Response · Authors · 2020-11-20
> **We added more experiments thanks to the reviewer's suggestion, and we are working on more.**
>
> First, we thank the reviewer for appreciating our result, for the high scoring, and for the very helpful suggestion.
>
> Q. I would be really interested in seeing the comparison of projective clustering vs simpler clustering methods such as k-means
>
> A: Added. This was a very good idea that we believe that significantly improved the paper. See graphs in section E.
>
>  The new graphs show that projective clustering can guarantee a better initialization, i.e., better accuracy before fine-tuning. This implies a fewer number of epochs.
>
> --------------------------------------------------------------------------------------------------------
>
> Q: Is projective clustering the only way to find clusters in multiple subspaces?
>
> A: It is a very good question and the answer depends on the definition of "clusters in multiple subspaces", i.e., the generative model. Projective Clustering aims to compute subspaces that maximize the likelihood, assuming that every point was generated by adding some noise to a point on a single subspace. As the reviewer suggests, we may assume soft clustering, e.g., that every point is a linear combination of points on a pair of subspaces. Such versions may be defined e.g. via Dictionary Learning. We expect that this paper will inspire many such generalizations in future works via our suggested network architecture or its variants.
>
> --------------------------------------------------------------------------------------------------------------------
>
> Q.The authors discussed extensions such as using error and distance, but no experiments were performed for the extensions. Some experiment results will be better to establish the flexibility of the framework of projective clustering in model compression tasks.
>
> A. We are doing our best to add such experiments, we believe we can provide some before the rebuttal period.

---

> ### Author Response · Authors · 2020-11-24
> **Rebuttal Reply #2**
>
> Dear Reviewer4,
>
> We are happy to update that we tried one of our suggested extensions as requested, specifically speaking we tried the first suggestion of $\ell^q$-error, with $q=1$.
>
> To compute a local minimum for the new cost function,  we still use EM algorithm. The only difference is that the SVD computation of the optimal subspace for a cluster of points ($k=1$) is replaced by an approximation algorithm for computing the subspace that minimizes sum over non-squared distances.
>
> However, since the deterministic approximation algorithms for the new problem ($\ell^1$ error) with $k=1$ take a time of $O(nd^4)$ (at least),  where $d=768$.  This change increased the running time of the algorithm from minutes to days (we need to run this approximation algorithm many times in the EM procedure.)
>
> For that and due to time constraints, we conducted our experiments only on one network (RoBERTA) on $2$ tasks from the GLUE benchmark (MRPC and RTE).
>
> Table 3 compares the original approach of the paper to the $\ell^1$ error, and it can be seen that mostly using the $\ell^1$ error as initialization is better than the $\ell^2$. However, for some reason (that needs further investigation), after fine-tuning for $2$ epochs both approaches reached almost the same accuracy, even more, the $\ell^2$ approach achieved a better accuracy sometimes. We leave this for future research.
>
> ------------------------------------------------------
>
> Finally, we would like to thank you again for appreciating our result, and for the very helpful suggestions to improve the paper.

---

> ### Author Response · Authors · 2020-11-24
> **Thank you**
>
> We appreciate the careful reading and thoughtful interchange, which aided us greatly in improving the clarity of our writing.

---

### Official Review · AnonReviewer1 · 2020-10-28
**Notitle**

**Rating:** 4
**Confidence:** 3

**Review:**

This paper extends the idea of using subspace clustering to compress the neural nets by considering multiple subspaces and projecting each point to its closest subspace. The paper needs more investigation on the related works. Basically, the idea and the technique is not novel. See the related literature below:
[1] Trittenbach, Holger, and Klemens Böhm. "One-Class Active Learning for Outlier Detection with Multiple Subspaces." Proceedings of the 28th ACM International Conference on Information and Knowledge Management. 2019.
[2] Liu, Risheng, et al. "Fixed-rank representation for unsupervised visual learning." 2012 IEEE Conference on Computer Vision and Pattern Recognition. IEEE, 2012.
[3] Xu, Dong, et al. "Concurrent subspaces analysis." 2005 IEEE Computer Society Conference on Computer Vision and Pattern Recognition (CVPR'05). Vol. 2. IEEE, 2005.
[4] Feng, Jianzhou, et al. "Learning dictionary via subspace segmentation for sparse representation." 2011 18th IEEE International Conference on Image Processing. IEEE, 2011.
---------------------------------------------
Pros:
•	Smoothly readable.
•	The contribution section is described thoroughly and properly.
•	Providing the codes for reproducing results.
---------------------------------------------
Cons:
Abstract:
•	The abbreviations like NLP or SVD should be defined first, then used.
•	Assuming that the reader already has corresponding field knowledge about systems such as GLUE, DistilBERT, or RoBERTa and mentioning them in the abstract may be bold.
•	Details of the methods such as the use of Aj matrix or k>1 subspace should not be mentioned in the abstract but rather in the contribution or introduction section accordingly.
•	The last sentence “Open Code for ….” Should not be mentioned in the abstract but in the code description section.
•	The figures 1-3 in the paper look not well organized, which makes the proposed simple idea to be extremely complex.
Results:
•	It would be better to discuss the comparable results more thoroughly.
•	Model compression literature should be reviewed and the typical methods should be compared with in the experiments.
Discussion and Conclusion:
•	Only discussion of the results is provided in this section and the conclusion is not provided explicitly.


Future Work:
•	Better not to start the section with numbered items right away. Better to have a starting sentence first.
Appendix B
•	Titled results before fine-tuning and includes figures with no explanation. Provide proper description and discussion for each subfigure.

---

> ### Author Response · Authors · 2020-11-20
> **Most of the writing issues are fixed as requested by the reviewer, and we are working on fixing the rest.**
>
> We thank the reviewer for the suggestions to improve the writing, and for the detailed review.
>
> Q: the idea and the technique is not novel. See the related literature below
>
> A: Of course, projective clustering is a classic old technique. We gave citations to all of the reviewer's suggestions.
> Nevertheless, as the title implies, our paper is the first that forges a link between projective clustering and Deep Learning.
> It also answers the question of how to apply this classic technique to deep learning, by presenting a new network architecture.
> The experimental results show that this new meeting of fields improves the results significantly in practice.
>
>  --------------------------------------------------------------------------------------------
>
> abstract:
>
> Q: The abbreviations like NLP or SVD should be defined first, then used.
>
> A: Fixed.
>
> Q: The last sentence “Open Code for ….” Should not be mentioned in the abstract but in the code description section.
>
> A. Fixed.
>
> As for the other comments about the abstract:
>
> All the questions of the reviewers regarding the abstract are answered in the introduction. We try to add more hints but due to the space constraints, we can not answer all the questions already in the abstract.
>
>  --------------------------------------------------------------------------------------------
>
> Results:
>
> Q It would be better to discuss the comparable results more thoroughly.
>
> A: Due to space limitations, such discussions appear in the appendix. More discussions will be added (before the rebuttal period ends) following the reviewer's request.
>
> Q: The conclusion is not provided explicitly.
>
> A: Fixed, thanks to the reviewer for pointing this out.
>
> Q:  Better not to start the section with numbered items right away. Better to have a starting sentence first. Appendix B
>
> A: Fixed.
>
>  Q:  Titled results before fine-tuning and includes figures with no explanation. Provide proper description and discussion for each subfigure.
>
> A: Fixed .

---

> ### Author Response · Authors · 2020-11-24
> **Rebuttal Reply #2**
>
> Dear Reviewer1
>
> We are happy to update you that we have added many experiments including (but not only) :
>
> 1.compressing fully connected layers using the suggested approach;  see section C.
>
> 2. Improving the accuracy of a given model using the new architecture while maintaining the same number of parameters; see Table 2
>
> We also updated the discussion, conclusion, and future work sections as requested.
>
> Finally, we would like to thank you again for your detailed comments and helpful review.

---

> ### Author Response · Authors · 2020-11-24
> **Paper Revised**
>
> We have taken care of all your comments. Please let us know in case we missed anything.

---

### Official Review · AnonReviewer5 · 2020-11-06
**Interesting work**

**Rating:** 5
**Confidence:** 3

**Review:**

This work proposes a new approach, based on projective clustering, for compressing the embedding layers of DNNs for natural language modeling tasks. The authors show that the trade-off between compression and model accuracy can be improved by considering a set of k subspaces rather than just a single subspace. Methods for compressing DNNs is an active area of research and this paper presents a promising approach to do so as well as interesting results.

Rating: The paper presents interesting ideas for compressing embedding layers. However, since this is an empirical paper, I would expect a more comprehensive set of empirical results and a better comparison with other related methods. Overall, the paper seems not very mature in its current form, hence my rating is 'Ok but not good enough - rejection'.

Pros
----
* The proposed method is appealing due to its simplicity and the idea of considering multiple subspaces for embedding is plausible in the context of compressing embedding matrices of NLP models.

* The results show improvements as compared to using just a single subspace.

* The framework provides several ideas for future works.

Cons
-----
* Typically, the SVD takes the form A = UDV, where U and V are the left and right singular vectors and the diagonal entries of D are the singular values. From the discussion it is not clear whether you factor the singular values into U, or whether you simply ignore the singular values? Also, how do you enforce the orthogonality constraints on U and V during the fine tuning stage? Have you considered a simpler low-rank factorization A = EF in your experiments, where no orthogonality constraints on E and F are imposed?

* It would be good to see the progression for k={2,3,4,5} in Figure 5 and 6. Further, the ensemble approach in Figure 6 hasn't been discussed in detail anywhere in the paper. It is not exactly clear to me how you are computing the ensemble.

* It would be very helpful to see some Tables that shows the total number of weights, accuracy, k, j, etc., in order to better understand the performance.

* How do you determine k and j in practice? Are you using some heuristic or are you simply doing a grid search?

* I would like to see how your method compares to ALBERT and whether a modified ALBERT (as you suggest in your future work section) is doing better.

* I would be interesting to see if you approach is also useful for compressing a fully connected layer in different settings. This should be easy to test and could be reported in the Appendix.

Minor comments:
--------------
* It is nice to see that you have many generalizations an extensions in mind, but this section appears very lengthy to me.

* compression rater -> compression rates

---

> ### Author Response · Authors · 2020-11-20
> **We thank the reviewer for the suggestions -- many experiments were added, and we are working on adding more.**
>
> First, we thank the reviewer for the detailed review and very helpful suggestions.
>
> Q: The paper presents interesting ideas for compressing embedding layers.
>
> A: We thank the reviewer for appreciating the ideas in our paper.
>
>  --------------------------------------------------------------------------------------------
> Q. I would expect a more comprehensive set of empirical results
>
> A: Following this reviewer's request,  many experiments were added. Including
>
>     1. compressing fully connected layers using the suggested approach.
>
>     2. checking how another clustering method can fit in our pipeline.
>
>     3. more results on other $k$-value as requested.
>
> Also, we are coding and running more experiments now -- hoping to finish before the deadline.
>
>  --------------------------------------------------------------------------------------------
>
> Q:  Typically, the SVD takes the form A = UDV...from the discussion, it is not clear whether you factor the singular values into U, or whether you simply ignore the singular values?
>
> A: As in previous papers, we use the more general factorization (as e.g. in NNMF) A=EF which corresponds to a pair of layers. This means that we did not ignore the singular values, and they can be assigned to either the left or right matrix. See next question.
>
>  --------------------------------------------------------------------------------------------
>
> Q: how do you enforce the orthogonality constraints on U and V during the fine-tuning stage?
>
> A: We did not.  This is a very good point that was added to the text. The orthogonalization is used to obtain a low rank approximation A~EF using SVD. From this point, we did not see an advantage to keep this property in the network.
>
>  --------------------------------------------------------------------------------------------
>
> Q: Have you considered a simpler low-rank factorization A = EF in your experiments, where no orthogonality constraints on E and F are imposed?
>
> A: This is exactly what we do after computing SVD, as explained in the previous answers. In fact, any non-orthogonal base to the span of the reduced matrix will do.  It is an interesting idea to try other non-orthogonal basis, as done e.g. in Dictionary learning.
>
>  --------------------------------------------------------------------------------------------
>
> Q:  It would be good to see the progression for k={2,3,4,5} in Figure 5 and 6.
>
> A: Added. We thank the reviewer for this good suggestion.
>
>  --------------------------------------------------------------------------------------------
>
> Q:  The ensemble approach in Figure 6 hasn't been discussed in detail anywhere in the paper.
>
> A:  Added. Due to space constraints, the graphs and explanations can be found in the appendix.
>
>  --------------------------------------------------------------------------------------------
>
>
> Q:  It would be very helpful to see some Tables that show the total number of weights, accuracy, k, j, etc., in order to better understand the performance.
>
> A: We thank the reviewer for the suggestion, and expect to finish this task before the end of the rebuttal.
>
>  --------------------------------------------------------------------------------------------
>
>
> Q: How do you determine k and j in practice? Are you using some heuristic or are you simply doing a grid search?
>
> A:  For a given compression rate $x$, we try multiple values of $k$ via binary search on $k$.,For every such $k$ value we compute the implied value  $j = (1-x)dn/(n+kd)$, and then check the compression result on those values.
>
>  --------------------------------------------------------------------------------------------
>
> Q: Apply your technique on ALBERT.
>
> A: We are implementing and running these experiments and expect to finish before the end of the rebuttal.
>
>  --------------------------------------------------------------------------------------------
>
> Q: Compress a fully connected layer in different settings using your approach.
>
> A: Added. See Appendix C for  LENET_300_100 and vgg19 models. The results are strong and better than the competitors as in the other experiments.
>
>  --------------------------------------------------------------------------------------------
>
> Q: It is nice to see that you have many generalizations and extensions in mind, but this section appears very lengthy to me.
>
> A: Fixed. We were indeed excited by our results and have many future ideas. However, most of this section was moved to the appendix.
>
>  --------------------------------------------------------------------------------------------
>
> Q: compression rater -> compression rates
>
> A: Fixed. We thank the reviewer for the careful reading.

---

> ### Author Response · Authors · 2020-11-24
> **Rebuttal Reply #2**
>
> Dear Reviewer5,
>
> We are happy to update the following:
>
> 1. We have added the suggested table, you can see it in the new version (Table 1). Please, feel free to ask for any change/improvement.
>
> 2. Experiments on Albert:
>
> We tested if the suggested architecture (MESSI) can improve the accuracy of a pre-trained model while maintaining the same number of parameters.  So we used the model ALBERT and factored its embedding layer to the suggested MESSI architecture using our pipeline.  Here, we choose the values of $k$ and $j$ such that the original embedding layer size is maintained (up to a very small change).  The results are so promising. Table 2 in the paper shows that we improved the accuracy of the original model for most of the tasks.
>
> 3. As previously updated, all other comments have been fixed, and experiments have been added.
> -------------------------------------------------
>
> Finally, we thank the reviewer for the comments that helped us further improve the quality of this paper.

---

> > ### Comment · AnonReviewer5 · 2020-11-24
> > **Thanks for the very detailed response**
> >
> > I would like to thank the authors for the very detailed response. All my questions have been addressed and I feel that the manuscript has improved quite a bit. The results in Table 1 and 2 are convincing. Based on the revision, I am more than willing to change my rating for this paper during the upcoming final discussion period.

---

> > > ### Author Response · Authors · 2020-11-24
> > > **Thank you**
> > >
> > > We are glad you like the updated paper.  Thank you for your careful and attentive reading, and we look forward to seeing your updated rating after our edits based on your comments.

---

### Decision · Program_Chairs · 2021-01-07
**Final Decision**

**Decision:**

Accept (Poster)

**Comment:**

The paper proposes to use projective clustering to compress the embedding layers of DNN. This is a novel interesting idea which can  impact the area of Knowledge distillation. There were some concerns about the empirical study which was addressed to some extent  by the authors during the rebuttal.